# Real-Time Surgical Problem Detection and Instrument Tracking in Cataract Surgery

**DOI:** 10.3390/jcm9123896

**Published:** 2020-11-30

**Authors:** Shoji Morita, Hitoshi Tabuchi, Hiroki Masumoto, Hirotaka Tanabe, Naotake Kamiura

**Affiliations:** 1Glory Ltd., 1-3-1 Shimoteno, Himeji-shi, Hyogo 670-8567, Japan; morita.shoji@mail.glory.co.jp; 2Graduate School of Engineering, University of Hyogo, 2167 Shosha, Himeji-shi, Hyogo 671-2280, Japan; kamiura@eng.u-hyogo.ac.jp; 3Department of Technology and Design Thinking for Medicine, Hiroshima University, 1-3-2 Kasumi Minami-ku, Hiroshima-shi, Hiroshima 734-8551, Japan; 4Department of Ophthalmology, Tsukazaki Hospital, 68-1 Waku, Aboshi-ku, Himeji-shi, Hyogo 671-1127, Japan; h.masumoto@tsukazaki-eye.net (H.M.); h.tanabe@tsukazaki-eye.net (H.T.)

**Keywords:** cataract surgery, neural networks, anomaly detection, image classification, image segmentation

## Abstract

Surgical skill levels of young ophthalmologists tend to be instinctively judged by ophthalmologists in practice, and hence a stable evaluation is not always made for a single ophthalmologist. Although it has been said that standardizing skill levels presents difficulty as surgical methods vary greatly, approaches based on machine learning seem to be promising for this objective. In this study, we propose a method for displaying the information necessary to quantify the surgical techniques of cataract surgery in real-time. The proposed method consists of two steps. First, we use InceptionV3, an image classification network, to extract important surgical phases and to detect surgical problems. Next, one of the segmentation networks, scSE-FC-DenseNet, is used to detect the cornea and the tip of the surgical instrument and the incisional site in the continuous curvilinear capsulorrhexis, a particularly important phase in cataract surgery. The first and second steps are evaluated in terms of the area under curve (i.e., AUC) of the figure of the true positive rate versus (1—false positive rate) and the intersection over union (i.e., IoU) obtained by the ground truth and prediction associated with the region of interest. As a result, in the first step, the network was able to detect surgical problems with an AUC of 0.97. In the second step, the detection rate of the cornea was 99.7% when the IoU was 0.8 or more, and the detection rates of the tips of the forceps and the incisional site were 86.9% and 94.9% when the IoU was 0.1 or more, respectively. It was thus expected that the proposed method is one of the basic techniques to achieve the standardization of surgical skill levels.

## 1. Introduction

It is known that there is a correlation between the number of cases performed and postoperative outcomes in surgery. For example, in gastric bypass surgery, the risk of postoperative complications is about twice as high with surgeons who have done less than 500 operations as with those who have done more than 500 operations [1]. In cataract surgery, the incidence of reactive corneal edema in the central corneal thickness at 2 hours after surgery was reported to be approximately 1.6 times greater for inexperienced surgeons than for experienced surgeons [2]. Therefore, shortening the acquisition time for surgical skills is one of the most important issues in medicine.

A common issue in surgical training is the difficulty in establishing objective criteria for the evaluation of surgical techniques. Although it has been pointed out that the quantitative measurement and standardization of surgical techniques is a necessary element for the systematic advancement of surgical training [3], there are a great variety of techniques in the continuous curvilinear capsulorrhexis (CCC) alone, which is essential in cataract surgery [4], and it is a difficult task to index all of them. Therefore, this study proposes a method for displaying the information necessary to quantify surgical techniques in real-time by identifying CCC and nuclear extraction, which have a high surgical difficulty, detecting the occurrence of surgical problems, and tracking cornea and surgical instruments.

In this study, we propose a method to determine whether CCC and nuclear extraction phases, which are critical surgical phases, have been successfully performed in cataract surgeries. The method’s capability to detect abnormality is evaluated by comparing the time when the proposed method detected a surgical problem with the actual time when the problem occurred. In addition, we propose a method for real-time detection of the corneal area, the tips of the forceps, and the incisional site of a patient in the CCC phase and evaluate its effectiveness.

## 2. Related Works

In recent years, video recording of surgical operations has become common practice and the use of video recordings in research pertaining to surgeries has become extremely popular. For surgical phase classification, we proposed a method [5] to extract the CCC and nuclear extraction phases, which are particularly important surgical phases in cataract surgery, with real-time utilization of one of the neural networks, InceptionV3 [6].

Detection of abnormal motions as well as detection and tracking of surgical instruments has also been actively conducted. Sakabe et al. [7] and Suzuki et al. [8] detected abnormal motions using video footage of the entire operating room. In Sakabe et al. [7], cubic higher-order local auto-correlation (CHLAC [9]) was used to detect features that are not normally visible, such as a scene where surgical instruments were picked up after being dropped on the floor. In Suzuki et al. [8], the study, focusing on the fact that the rapid movement seen in the video occurs when there are problems in operations, detected abnormal motions based on the amount of change in the video file size. However, these detection methods are difficult to apply to cataract surgeries which involve less movement, and it is not possible to detect surgical problems before they occur.

In the detection and tracking of surgical instruments, detection methods using bounding boxes [10] and segmentation methods [11] have been proposed. However, it is difficult to track the fine movements of the tips of surgical instruments, although the methods are suitable for tracking the rough movements of surgical instruments. Furthermore, surgical evaluation based on the positional relationship between the affected area and instruments is required, but the above method of tracking the instruments alone cannot be used for such an evaluation. A study that evaluated the technology of robotic surgery has been reported [12]; however, the proposed method is not applicable to cataract surgery performed manually by surgeons because the motion of the surgical instruments is detected and evaluated by a gyroscope attached to a robot.

Marco et al. [13,14] proposed a real-time simulation system for cataract surgery based on virtual reality technology. They developed Phaco-emulsification [13], using a mesh-less shape-based dynamic simulation algorithm and a smoothed particle hydrodynamics-based scheme. On the other hand, the system [14] was developed on the premises of using a three-dimensional tactile device and binocular display. Cataract surgery training is comprehensively available not only for Phaco but also for CCC, by applying the said system [14]. The systems [13,14] are clearly useful in acquiring numerical data associated with track dislocation, speed change, and so forth, of surgical instruments because they occur in simulations on computers. In other words, the systems [13,14] will be powerful devices to evaluate the surgical techniques of cataract surgery. In actual cataract surgery, however, acquiring the above numerical data is not as simple a task as the data acquisition performed on simulation. One of the objectives of the proposed method is to acquire the numerical data of various regions of interest from actual cataract surgery. If on-line data acquisition is feasible during actual cataract surgery, it is expected that the evaluation established by the systems [13,14] is applicable to actual surgical techniques. Besides, it is probable that the effectiveness of surgical training on simulation is quantized with the proposed method.

## 3. Datasets

Ophthalmologists working at Saneikai Tsukazaki Hospital (Himeji, Japan) watched video recordings of cataract surgeries performed at the hospital, and checked the time points when CCC started, nuclear extraction finished, and surgical problems occurred. Such time points were registered on electronic files. The use of these video recordings has been approved by the Ethics Committee of Tsukazaki Hospital.

Let us discuss surgical problem detection. In this detection, 425 video recordings of cataract surgeries were used. The resolution of the videos was 1920 × 1080 at a frame rate of 30 FPS with a mean duration of about 1018 s and a standard deviation (SD) of about 1046 s. The above electronic files were used to annotate surgical phases in the videos using labels. The number of classes is two. In other words, the label “important phase” is assigned to frames corresponding to the period between the two time points, from the time when CCC started through when nuclear extraction finished, while the label “other” is assigned to frames corresponding to the period exclusively included as time slots when neither CCC nor nuclear extraction were performed. The mean (SD) duration between the start of CCC and the end of the nuclear extraction was about 376 s (427 s), while the mean (SD) duration of the other phases was 642 s (798 s).

In surgical problem detection, we evaluated the neural network’s performance using 310 training data (57 with problems and 253 without problems), 15 validation data (5 with problems and 10 without problems), and 100 test data (50 with problems and 50 without problems) from 425 videos. The video frame rate was downsampled to 1 FPS, and the resolution was downsampled to 256 × 168 to perform surgical phase recognition and problem detection frame-by-frame. This yielded 422,559 images from 425 videos.

Recall that the above electronic files have information on the occurrence of surgical problems. In addition to one of the two labels, important phase or other phase, a label to indicate the presence or absence of a surgical problem was also added to each video frame. The labeling was performed based on information associated with surgical problems in the above files, and it is safely said that the annotation of surgical problems was made by ophthalmologists working at Tsukazaki Hospital. The phase breakdown of the obtained image data is shown in Table 1, and a sample of the actual images of each phase are shown in Figure 1. Additionally, a problem breakdown is tabulated in Table 2.

Also, the performance of detecting the surgical instrument was only evaluated for the CCC phase without problems. The occurrence of a problem in cataract surgery often results in either a low rate of progress or in interruption. The surgery then tends to consume much time. In such cases, no remarkable change appears between the consecutive frames of the video in which the surgery is recorded. When employing machine learning, acquiring data with high diversity is preferable to acquiring a large amount of data with low diversity. In other words, training data used to construct discrimination models by machine learning should be prepared from frames with clear changes made by surgical instruments. This is why surgical instrument detection is only applied to cases where no problems occur. Of the 302 videos, 211 videos were utilized as training data, 30 as validation data, and 61 as test data. As with surgical problem detection, the video frame rate was downsampled to 1 FPS, and the resolution was downsampled to 256 × 128 to perform surgical instrument detection. This resulted in 9354 training data, 981 validation data, and 2299 test data from 302 videos, for a total of 12,634 images. Ophthalmologists working at Tsukazaki Hospital annotated surgical instruments in videos with labels. They then used the annotation tool known as LabelMe [15]. The labels are as follows: the patient’s corneal area, the tips of the forceps, and the incisional site. Examples of an input image and the corresponding segmentation images are shown in Figure 2.

## 4. Neural Networks (NN)

### 4.1. InceptionV3

In this study, we used a convolutional neural network (hereinafter referred to as NN) model, known as InceptionV3, to recognize the cataract surgery phases and detect surgical problems. It is considered that surgery-phase recognition deeply depends on NN’s capability in classifying the given data. Surgical-problem detection requires extremely short response times for the data presented to the trained NN. It can generally be considered that data-classification accuracy and short response time are related to the transactions. We selected InceptionV3, which provides high performance with real-time operating capability, by referring to the benchmark [16] which investigated the relationship between the performance of major NN and computational capacity.

InceptionV3 replaces *n* × *n* convolution with 1 × *n* and *n* × 1 convolution called the Inception module and introduces a mechanism to reduce the number of channels in 1 × 1 convolution called the bottleneck layer, which reduces the number of parameters and computation time and suppresses the vanishing gradient.

The number of parameters and the number of computations in the convolutional layer is represented by the Equations (1) and (2), respectively, where $k_{h}$, $k_{w}$, h, w, $c_{in}$, and $c_{out}$ are the vertical kernel size of the convolution, the horizontal kernel size of the convolution, the vertical size of the input tensor, the horizontal size of the input tensor, the channel size of the input tensor, and the channel size of the output tensor, respectively.
(1)$$\left(\mathrm{Number}\;\mathrm{of}\;\mathrm{parameters}\right)=k_{h}k_{w}c_{in}c_{out}+c_{out}\;$$
(2)$$\left(\mathrm{Number}\;\mathrm{of}\;\mathrm{computations}\right)=k_{h}k_{w}hwc_{in}c_{out}\;$$

Therefore, if *n* × *n* convolution is replaced by 1 × *n* and *n* × 1 convolution, the number of parameters and the number of computations can be reduced to about *n*/2. In addition, the bottleneck layer can significantly reduce the number of parameters and the number of computations thereafter by reducing the channel size of the tensor with a small number of parameters and computation times.

In this study, we used five types of Inception modules, as shown in Figure 3. “Base” is the input tensor to the Inception module. The “conv” is a convolutional layer, where the convolution is performed on a local region of the image and learns high-order features of the image. The pooling layer compresses the tensors to reduce the amount of computation and prevent over-learning. “Max Pooling” computes the maximum value of the local region of the tensor, and “Average Pooling” computes the average value and compresses it. “Filter Concat” represents the concatenation of tensors.

### 4.2. scSE-FC-DenseNet

In this study, one of the segmentation NNs, scSE-FC-DenseNet, was used to detect the corneal area of the patient and track the surgical instruments. High computational complexity is imposed on a segmentation NN, and hence the NN must be carefully designed if its response time is shortened so that the NN can safely be said to be nearly real time. The scSE-FC-DenseNet is an FC-DenseNet [17], which incorporates the Dense block proposed in DenseNet [18] into U-Net [19], with an attention mechanism called the scSE (Spatial and Channel Squeeze & Excitation) module [20]. It is discussed in [17] that employing DenseNet enables us to easily adjust the computational complexity imposed on the NN.

U-Net is a type of Fully Convolutional Networks (FCNs). As shown in Figure 4, it is characterized by the introduction of a mechanism called “skipped connections” that utilizes information during the encoding process when the encoded image is decoded. By using the skip connection, small features that are lost due to compression of the image by the pooling layer can be restored.

DenseNet is a NN that uses a structure called Dense block, which combines a skipped connection and a bottleneck layer. This structure allows us to construct NNs that do not increase the number of parameters explosively, even when the convolutional layer is more multi-layered. FC-DenseNet is a NN obtained by replacing the usual convolutional layer used in U-Net with this Dense block.

The scSE module is a combination of the Spatial Squeeze and Excitation (cSE) proposed in SE-Net, an image classification NN [21], which averages the whole image for each channel (Squeeze) and gives its attention (Excitation), and an sSE (Channel Squeeze and Spatial Excitation) that squeezes to the channel direction and excites each pixel. This module can effectively introduce the attention mechanism in a segmentation NN.

In this study, we used the scSE-FC-DenseNet40, which consists of 40 layers of Dense blocks, with a scSE module embedded after each Dense block.

## 5. Monitoring of Cataract Surgery

### 5.1. Problem Detection in Critical Phases

For the problem detection in the critical phases of cataract surgery, the convolutional NN model known as InceptionV3 was used to recognize two surgical phases, CCC to nuclear extraction and others. The input to the NN model was 299 × 168 × 3 color images. The number of output layer neurons was set to 2, the number of surgical processes to be recognized. The class corresponding to the maximum value of the output layer neurons was set to the surgical phase estimated by the model. The structure of the InceptionV3 model used in the study is shown in Table 3. The “type”, “Patch size/stride”, and “Input shape” columns indicate the type of layer, the size of the local window and its stride width, and the size of the tensor to be input to each layer, respectively.

The NN model was trained by initializing each parameter with trained parameters in the ILSVRC 2012 dataset [22]. The training parameters were set to a batch size of 32, the loss function of categorical cross-entropy, the optimization function of momentum SGD (learning rate, 0.0001; momentum, 0.9), and the number of epochs of a maximum of 300. In addition, for pre-processing, the pixel values of the images were normalized in the range of 0 to 1. To prevent over-learning, we randomly applied the image augmentation process as shown in Table 4. To address the imbalance in the number of images for each phase, the classes with a small number of images were trained multiple times within one epoch. The network was trained on a system with two NVIDIA GTX 1080 Ti GPUs and the evaluation was done on a single GPU.

We then used a similar method to identify whether any problems occurred during the critical phases of surgery. The network, training parameters, and image augmentation process parameters used were the same as for the critical phase recognition.

Next, our two NN models were used to detect surgical problems and estimate the time of their occurrence. The moving average of the output values of InceptionV3 was used to stabilize the output results of the phase recognition and problem detection. We set the number of frames for moving average to 10 because increasing the number of image frames used for moving average slows down the response time. First, the images obtained from the surgical videos were arranged in chronological order and inputted into the InceptionV3 to obtain the output results. In the output layer, three neurons correspond to three classes, i.e., CCC, nuclear extraction, and others. Let $eval_{i^{j}}$ denote the value of the jth output neuron at the ith second, the value of the moving average is denoted as follows.
(3)$$Ave_{i^{j}}=\frac{1}{10}\overset{i}{\underset{k=i-9}{\sum }}eval_{k^{j}}\;(i>9,\;1\leq j\leq 2)\;$$

Next, the risk of surgical problems, $D_{t}$, was defined and calculated by the Equation (4), where $PA_{i^{j}}$, $TA_{i^{j}}$, i, and j are the moving average of the output values of the surgical phase recognition NN calculated by the Equation (3), the moving average of the output values of the surgical problem detection NN calculated by the Equation (3), the ith second of the movie, and the jth output neuron, respectively.
(4)$$D_{t}=PA_{i^{j}}TA_{i^{j}}\;\left(i>9,\;1\leq j\leq 2\right)\;$$

Finally, the time when the risk $D_{t}$ exceeded the threshold was set as the time of the surgical problem estimated by the NN.

### 5.2. Tracking Cornea and Surgical Instruments during the CCC Phase

In this study, scSE-FC-DenseNet40, a segmentation NN built with 40 layers of Dense block, was used to detect the corneal area, the incisional site, and the tips of surgical instruments during the CCC phase. The 256 × 128 color images were used as the input and 256 × 128 × 3 as the output for NN (input image size, 256 × 128; the number of classes, 3). The actual NN structure used in the study is shown in Table 5. The “Skip Connection” column indicates the connection between the layers, which means that the output tensor of the layer corresponding to Output (*x*) is connected to the input tensor of the layer corresponding to Concat (*x*). The training parameters were set to a batch size of 16, the loss function of mean square error, the optimization function of AdaBound (learning rate: 0.001) [24], and the number of epochs of a maximum of 300. In addition, for pre-processing, the pixel values of the images were normalized in the range of 0 to 1. To prevent over-learning, we randomly applied the image augmentation process as shown in Table 6. The network was trained on a system with two NVIDIA GTX 1080 Ti GPUs and the evaluation was done on a single GPU.

## 6. Experimental Results

### 6.1. Problem Detection in Critical Phases of Cataract Surgery

The results of the frame-by-frame recognitions of critical phases and surgical problems using InceptionV3 are shown in Table 7 and Table 8. The correct response rates for the critical phases and others were 84.4% and 94.9%, respectively, with a mean correct response rate of 91.3% (Table 7). The correct response rates for “without problems” and “with problems” were 86.0% and 91.2%, respectively, with a mean correct response rate of 90.2% (Table 8). Note that the results were obtained by applying the proposed method to videos with frames in which surgical instruments appeared. The proposed phase recognition and problem detection can be applied to a video consisting of consecutive frames without surgical instruments, because in addition to the instruments, the eye area is also targeted for training. However, their accuracy decreases in such cases.

Next, we calculated the risk level $D_{t}$ for each video using Equation (4). Based on the obtained risk levels, the videos without problems were labeled as negative, and the videos with problems as positive, and a ROC curve was drawn (Figure 5). Table 9 shows the results of the surgical problem detection for each video when using the obtained threshold value as a reference. The AUC was 0.970, and the correct response rates were 94% for “without problems” and 90% for “with problems”, with a mean correct response rate of 92%. The histogram in Figure 6 represents the differences between the time when the risk level $D_{t}$ exceeded the threshold and the time when the ophthalmologist determined that a problem occurred in a video, for the videos correctly recognized as “with problems”. The “0” time point on the horizontal axis means that there was no difference between the problem-occurring time determined by the ophthalmologist and that detected by the NN. It is shown that the NN detected problems earlier than the ophthalmologist in 42 out of 44 cases. Figure 7 shows examples of the risk level $D_{t}$ for videos without problems Figure 7A and videos with problems Figure 7B. The figure clearly shows that the risk level in Figure 7B is larger than Figure 7A.

### 6.2. Tracking Cornea and Surgical Instruments during the CCC Phase

The tracking of the cornea and surgical instruments was evaluated with the Equation (5) and $ACC_{IoU\geq N}$, defined based on the IoU (Intersection over Union) shown in Figure 8. $ACC_{IoU\geq N}$ is denoted by the Equation (6).
(5)$$\mathrm{IoU}=\frac{Area\;of\;Overlap}{Area\;of\;Union}\;$$
(6)$$ACC_{\mathrm{IoU}\geq N}=\frac{Number\;of\;\mathrm{data}\;\mathrm{satisfying}\;\mathrm{IoU}\geq N\;data}{Number\;of\;all\;data}\;$$

The correct response rates of detecting the cornea, the incisional site, and the tips of the forceps are shown in Table 10. Note that the value of *N* is a parameter related to IoU. IoU takes the value 1 as its maximum. IoU = 1 means the prediction result obtained by the proposed method perfectly matches the ground truth. IoU ≥ *N* means that the ratio of the area obtained by overlapping the prediction result with ground truth compared with the area obtained by uniting the former to the latter is larger than or equal to *N*. The correct response rate for the cornea showed remarkably high accuracy. On the other hand, the correct response rates for the incisional site and the tips of the forceps were 94.9% and 86.9% respectively, when the value of *N* in the Equation (6) was 0.1. In other words, the value obtained with the Equation (6) is not high under *N* = 0.8. However, as shown in Figure 2, the area for which the correct label is assigned for surgical instrument detection is exceedingly small; therefore, even with the criteria, it seems that results for tracing the incisional site and the tips of forceps causes no major troubles during practical use. It can be thus considered that the NN is capable of tracking instruments successfully. Figure 9 shows examples of segmentation results and the visualization of the segmentation. The image Figure 9A is an input image, and the images Figure 9B–D represent the segmentation result of the cornea, the tips of the forceps, and the incisional site, respectively. The image Figure 9E shows a visualization of the cornea (red circle), the tips of the forceps (blue circle), and the incisional site (green circle). The trained scSE-FC-DenseNet40 outputs a value belonging to the range 0 to 1 as certainty values of segmentation for each pixel. It is judged that the segmentation result is appropriate as its certainty approaches the value 1. The certainty values of segmentation for each pixel are made into a gradation map. The maps overlap with images to be segmented as shown in Figure 9B–D. Here, the color becomes bluer (or redder) as certainty of segmentation for each pixel approaches the value 0 (or 1). Binarization is executed provided that the threshold certainty of segmentation for each pixel is set to the value 0.5. The circumscribed circle is next depicted for each of the segmented objects. Figure 9E is then obtained as a result. Note that its ground truth is depicted as shown in Figure 2.

## 7. Discussion

First, the validity of estimating the problem-occurring time by identifying surgical problems is examined. One of the possible applications of this problem detection NN is the construction of a system that detects a problem during surgery and alerts physicians to stop the surgery. This application considers the fact that the NN could estimate the occurrence of a problem earlier than an ophthalmologist in most cases. This was shown in the histogram in Figure 6. This implies that the risk alert based on the proposed method is quite promising. However, the proposed method estimates the problem-occurring time more than one minute before the ophthalmologist in most videos. The estimated time by the NN may not be consistent with the estimated time that ophthalmologists prefer. For example, residents tend to make larger incisions than experienced ophthalmologists during the CCC phase, and the proposed method may place too much weight on this tendency. It seems that the above are caused by differences that sometimes arise in evaluation criteria for surgical techniques between experienced ophthalmologists and the proposed method. The proposed method should be based on the evaluation criteria for surgical techniques that the majority of experienced ophthalmologists consider to be appropriate. To achieve this objective, it is unfavorable that differences sometimes arise in the evaluation criteria. The proposed method therefore must be refined so that its criteria can be as close as possible to those of experienced ophthalmologists. Refining the proposed method thus makes it possible to create a practical problem detection system that can estimate a problem based on causal relationships and which demonstrates greater consistency with ophthalmologists’ judgement criteria.

Next, we discuss the possible cause that led to a lower detection accuracy for the tips of the forceps compared with that for the incisional site in the tracking of the cornea and surgical instruments during the CCC phase. The incisional site is the border between the outer and middle of the cornea and is easy to locate, while the tips of the forceps are inside the cornea, which can make it difficult to see depending on the condition of the eye, as shown in Figure 10. This may have resulted in a lower detection accuracy for the forceps than for the incisional site. To improve accuracy, we have tried to apply several schemes of general image processing to make it easier for the NN to recognize the eye and instruments as a preprocessing step. An essential scheme has yet to be discovered, however. It is possible that engaging in schemes specialized for cataract surgery is preferable to examining well-known schemes. We will continue to search for such an image processing scheme.

Cataract is one of the most frequent causes of blindness, especially in poor countries. To reduce the number of blindness due to cataract, education of ophthalmologists performing cataract surgeries is of absolutely importance all over the world. Eyesi Surgical, a virtual reality simulator for intraocular surgery training, has been developed in Germany [25]. On the other hand, femtosecond laser–assisted cataract surgery [26] has been receiving a lot of attention. These high-end machines are unavailable for ophthalmologists working in developing countries. Developing the AI-supported system based on the proposed method will make it possible to remarkably reduce the costs for education of ophthalmologists and to enhance its convenience. In addition, the proposed method seems to be useful in grading surgical skill levels with marks (e.g. Appendix A). In other words, it is comparatively easy to achieve the skill-level standardization with the proposed method. Experienced ophthalmologists can also optimize their surgical guidance for inexperienced ophthalmologists, while referring to evaluation reports submitted by the proposed method.

## 8. Conclusions

In this study, we proposed a cataract surgery monitoring system using the NN. This system consists of two steps, and the step-by-step instructions are as follows. In the first step, the extraction of the CCC and nuclear extraction phases, which are critical surgical phases in cataract surgery, is performed using a convolutional NN called the InceptionV3. The surgical videos are downsampled to a frame rate of 1 FPS and a resolution of 299 × 168 to decompose into frames. Using these frames, the critical phase recognition (the CCC to nuclear extraction) and the surgical problem detection in critical phases are performed using the InceptionV3. Next, the frames from a movie are arranged in chronological order, and the 10-s moving average of these two models is calculated, and the obtained values are multiplied to calculate the risk level $D_{t}$. Finally, a ROC curve is drawn based on the risk level value of $D_{t}$ for each video and evaluated with the AUC. In our experiment, the accuracies of the critical phase recognition and the problem detection in critical phases were 91.3% and 90.2%, respectively, and the AUC for each video was 0.970.

In the second step, the tracking of the cornea and surgical instruments during the CCC phase is performed using scSE-FC-DenseNet40, a segmentation NN. The proposed method uses scSE-FC-DenseNet40, which consists of 40 layers of Dense blocks that incorporate the scSE module into FC-DenstNet. The scSE module is an attention mechanism proposed for the segmentation NN. In our experiment, the results were as follows: $ACC_{IoU\geq 0.8}$ = 99.7% for the cornea, $ACC_{IoU\geq 0.1}$ = 86.9% for the tips of the forceps, and $ACC_{IoU\geq 0.1}$ = 94.9% for the incisional site.

Further steps needed to improve the system include matching the judgment criteria for surgical problems between the NN and the ophthalmologist by performing problem detection with the information on the cornea and instrument tracking, as well as performing the cornea and instrument tracking during the nuclear extraction phase.

## Figures and Tables

**Figure 1 jcm-09-03896-f001:**
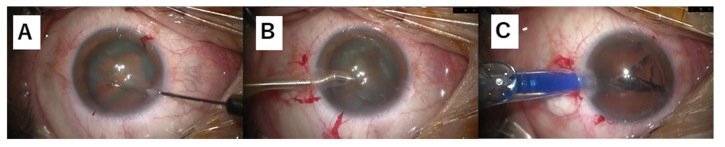
Sample images of each surgical phase. (**A**) CCC (Inamura forceps, retro illumination method), (**B**) nuclear extraction, (**C**) others (intraocular lens insertion).

**Figure 2 jcm-09-03896-f002:**
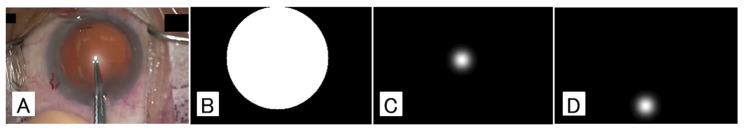
Segmentation example of surgical instrument detection. (**A**) input image, (**B**) ground truth of corneal area, (**C**) ground truth of forceps’ tips, (**D**) ground truth of incisional site.

**Figure 3 jcm-09-03896-f003:**
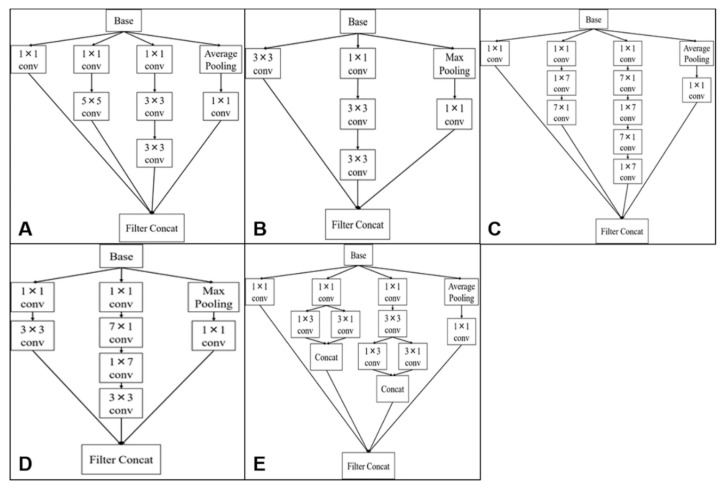
The Inception modules used. “Base” means the input tensor to the Inception module. The “conv” means a convolutional layer at which the convolution is performed for a local region of the image, and learns high-order features of the image. The pooling layer compresses the tensors to reduce the amount of computation and prevent over-learning. “Max Pooling” computes the maximum value of the local region of the tensor, and “Average Pooling” computes the average value and compresses it. “Filter Concat” represents the concatenation of tensors. (**A**) The module obtained by replacing part of the 5 × 5 convolution with 3 × 3 convolutions. (**B**) The module obtained by replacing 5 × 5 convolution with 3 × 3 convolutions. (**C**) The module obtained by replacing 7 × 7 convolution with 1 × 7 convolution and 7 × 1 convolution. (**D**) The module obtained by replacing 7 × 7 convolution with 1 × 7 convolution, 7 × 1 convolution and 3 × 3 convolution. (**E**) The module obtained by replacing 3 × 3 convolution with 1 × 3 convolution and 3 × 1 convolution.

**Figure 4 jcm-09-03896-f004:**
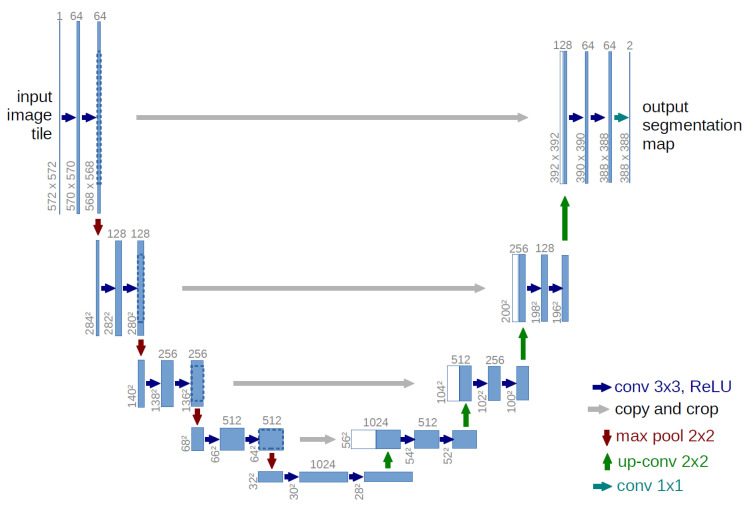
Network structure of U-Net, taken from [19].

**Figure 5 jcm-09-03896-f005:**
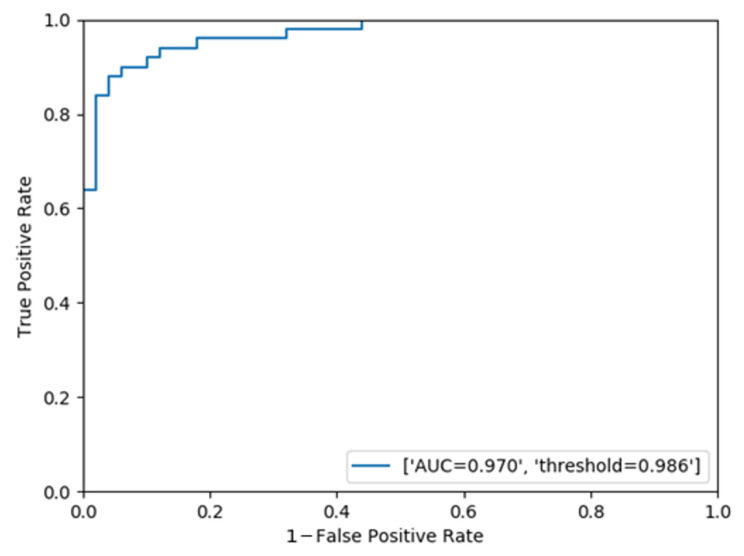
ROC curve of surgical problem detection per video. This evaluation is based on the risk level $D_{t}$.

**Figure 6 jcm-09-03896-f006:**
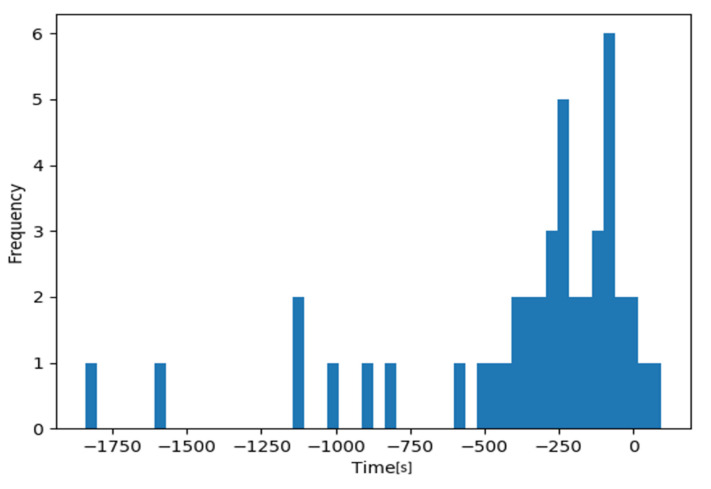
Differences between the problem-occurring time detected by the NN and the problem-occurring time determined by the ophthalmologist. The “0” time point on the horizontal axis means that there was no difference between the problem-occurring time determined by the ophthalmologist and that detected by the NN. If the time is negative, it means that the NN detected the problem earlier than the ophthalmologist.

**Figure 7 jcm-09-03896-f007:**
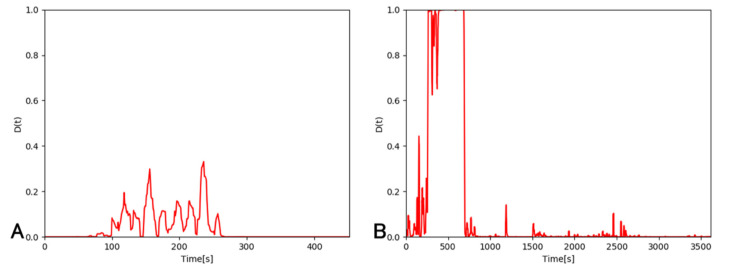
Difference in risk levels for videos with or without problems. (**A**) the risk level for videos without problems, (**B**) the risk level for videos with problems.

**Figure 8 jcm-09-03896-f008:**
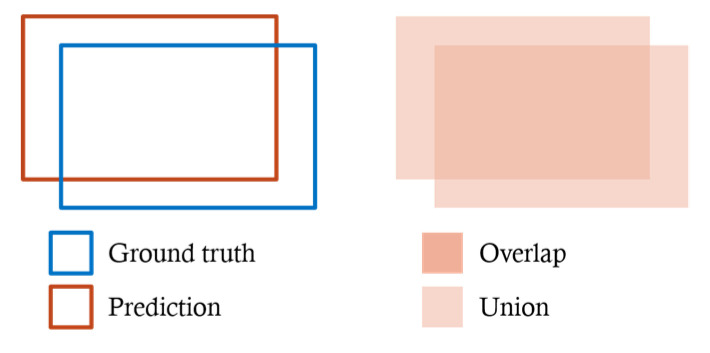
Overlap and Union regions used to calculate the IoU. IoU is specified by the ground truth and prediction associated with the region of interest, and takes a value in the range of 0 to 1.

**Figure 9 jcm-09-03896-f009:**
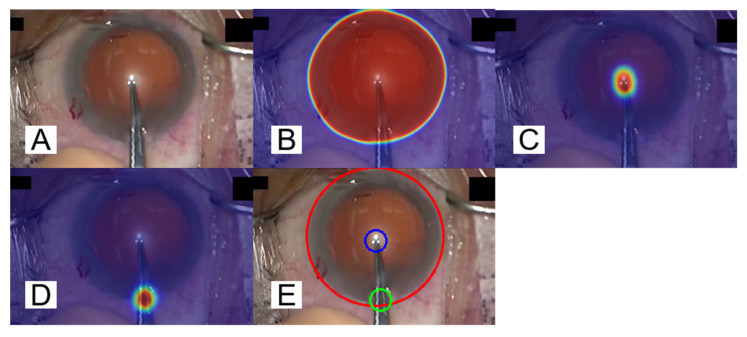
Sample images of a patient’s corneal area and the visualization of surgical instrument detection results. NN outputs the certainty of segmentation for each pixel in the range of 0 to 1. The certainty that is closer to 1 indicates a class that NN wants to detect. Images (**B**–**D**) are overlapping image (**A**) with gradation maps, showing blue when the certainty value is 0, and red when the certainty value is 1. Image (**E**) shows a circumscribed circle of the segmented figure by binarizing with the certainty of 0.5 as a threshold value. In addition, the ground truth is the same as in Figure 2.

**Figure 10 jcm-09-03896-f010:**
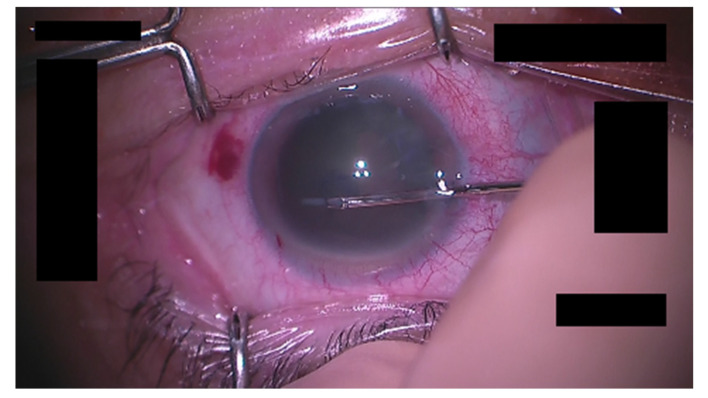
Example image of a hard-to-see forceps tips.

**Table 1 jcm-09-03896-t001:** Breakdown of datasets for the detection of cataract surgical phases.

Detected Class	Training Data (Images)	Validation Data (Images)	Test Data (images)	Total (Images)
CCC to nuclear extraction (normal)	60,304	3492	10,085	73,881
CCC to nuclear extraction (abnormal)	37,532	2561	42,196	82,289
Others	153,086	11,759	101,544	266,389
Total	250,922	17,812	153,825	422,559

**Table 2 jcm-09-03896-t002:** Breakdowns of surgical problems. As there were surgeries in which multiple problems occurred at the same time, the number of surgical videos with problems does not correspond to the total number of problem breakdowns.

Type of Problem	Number of Events
Vitreous prolapse	44
Capsule rupture	39
Damage to the iris	31
Iris prolapse	30
Rupture of the zonule of the Zinn	15
Dropped nucleus	14
Discontinuous CCC	13
CCC tear	7
Wound suture	5

**Table 3 jcm-09-03896-t003:** The InceptionV3 model used.

Type	Patch Size/Stride	Input Shape
Convolution	3 × 3/2	299 × 168 × 3
Convolution	3 × 3/1	149 × 83 × 32
Convolution padded	3 × 3/1	147 × 81 × 32
Max pooling	3 × 3/2	147 × 81 × 64
Convolution	1 × 1/1	73 × 40 × 64
Convolution	3 × 3/1	73 × 40 × 80
Max pooling	3 × 3/2	71 × 38 × 192
Inception	As in Figure 3A	35 × 18 × 192
2 × Inception	As in Figure 3A	35 × 18 × 256
Inception	As in Figure 3B	35 × 18 × 288
4 × Inception	As in Figure 3C	35 × 18 × 288
Inception	As in Figure 3D	17 × 8 × 768
2 × Inception	As in Figure 3E	8 × 3 × 1280
Average pooling	8 × 3	8 × 3 × 2048
Full connection		2048
Full connection		1024
SoftMax		2

**Table 4 jcm-09-03896-t004:** Types of image processing used for learning surgical phase recognition and their parameters.

Types	Parameters
Rotation	Up to 90 degrees
Horizontal movement	Up to 20%
Vertical movement	Up to 20%
Shear conversion	Up to 5 degrees
Scaling	Up to 10%
Channel shift	Up to 100
Flip horizontally	
Flip vertically	
Random erasing [23]	Up to 25%

**Table 5 jcm-09-03896-t005:** Structure of the scSE-FC-DEnseNet40 used.

Type	Patch Size/Stride	Input Shape	Skip Connection
Convolution	3 × 3/1	256 × 128 × 3	
2 × (Dense block & scSE module)		256 × 128 × 48	
Convolution	1 × 1/2	256 × 128 × 80	
4 × (Dense block & scSE module)		128 × 64 × 80	Output (1)
Convolution	1 × 1/2	128 × 64 × 144	
8 × (Dense block & scSE module)		64 × 32 × 144	Output (2)
Convolution	1 × 1/2	64 × 32 × 272	
6 × (Dense block & scSE module)		32 × 16 × 272	Output (3)
Transposed Convolution	3 × 3/2	32 × 16 × 96	
8 × (Dense block & scSE module)		64 × 32 × 368	Concat (3)
Transposed Convolution	3 × 3/2	64 × 32 × 128	
4 × (Dense block & scSE module)		128 × 64 × 272	Concat (2)
Transposed Convolution	3 × 3/2	256 × 128 × 64	
2 × (Dense block & scSE module)		256 × 128 × 144	Concat (1)
Convolution	1 × 1/1	256 × 128 × 176	
Sigmoid		256 × 128 × 3	

**Table 6 jcm-09-03896-t006:** Types of image processing used for learning corneal and positions of surgical instruments, and their parameters.

Types	Parameters
Rotation	Up to 90 degrees
Horizontal movement	Up to 20%
Vertical movement	Up to 20%
Shear conversion	Up to 5 degrees
Scaling	Up to 20%
Flip horizontally	
Flip vertically	

**Table 7 jcm-09-03896-t007:** The results of critical phase recognition per image using the NN.

	Detected Class	Critical [%]	Others [%]	Correct Response Rate [%]
Correct Class				
Critical		84.4	15.6	84.4
Others		5.1	94.9	94.9
				Mean: 91.3%

**Table 8 jcm-09-03896-t008:** The results of problem detection per frame using the NN.

	Detected Class	Without Problems (%)	With Problems (%)	Correct Response Rate (%)
Correct Class				
Without problems		86.0	14.0	86.0
With problems		8.8	91.2	91.2
				Mean: 90.2%

**Table 9 jcm-09-03896-t009:** The results of problem detection per video using the NN.

	Detected Class	Without Problems (%)	With Problems (%)	Correct Response Rate (%)
Correct Class				
Without problems		94	6	94
With problems		10	90	90
				Mean: 92%

**Table 10 jcm-09-03896-t010:** Results of cornea and surgical instrument detections during the CCC phase. *N* is a parameter related to IoU. IoU ≥ *N* means that the ratio of area obtained by overlapping the prediction result with ground truth compared with the area obtained by uniting the former to the latter is larger than or equal to *N*.

	N	0.1	0.2	0.3	0.4	0.5	0.6	0.7	0.8	0.9
$ACC_{IoU\geq N}$										
Cornea		99.9	99.9	99.9	99.9	99.9	99.9	99.9	99.7	92.6
Incisional site		94.9	44.7	7.85	7.66	7.66	7.66	7.66	7.66	7.66
Tips of forceps		86.9	28.5	6.14	6.14	6.14	6.14	6.14	6.14	6.14

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
