# Peer review of "Real-Time Surgical Problem Detection and Instrument Tracking in Cataract Surgery"

_jcm, 2020, doi:10.3390/jcm9123896_

Round 1
Reviewer 1 Report
The manuscript describes a technology for analysis of capsulhorexis videos through modern machine learning technologies, in order to extract information about corneal incisions and surgical tools.
The paper is technically sound and well organized. Even if the used technologies are not novel per-se, the customization and the model architecture appears convincing and well shaped.
Presented results are technically convincing, but the evaluation
part is lacking.
After the devlopment of such a sophisticated technology for tracking surgical tools, I would expect that the data could be used for performing more sophisticated domain-based visual analysis.
There are many examples of possible applications:
1. evaluation of correctness of movements and mapping to intervention outcomes
2. visualization of trajectories (in 3D or 2D)
3. quantification of data and dimension reduction for more sophisticated data analysis
Since all these kind of evaluations are missing, the contributions of the paper appear somehow limited from the medical point of view, and I cannot recommend for publication.
Additional comments:
The first sentence in the abstract is not clear:
It has been pointed out that the quantitative evaluation of surgical
the technique is a necessary element for the systematic promotion
of surgical education, but it is difficult to standardize all of
them because of the various methods used.
In general what are the main motivations, and the actual contributions
of the paper? It is not really clear from the abstract.
I also don't like to read in the abstract quantitative measures without
a context and references.
I think also that the organization of the paper would improve
if the discussion of related work is separated from the introduction.
To this end, it would be useful to add a discussion about surgical training
in cataract extraction. For an overview, authors can consider the papers:
Real Time Simulation of Phaco-emulsification for Cataract Surgery Training.
M Agus, E Gobbetti, G Pintore, G Zanetti, A Zorcolo
VRIPHYS, 91-100
Real-time Cataract Surgery Simulation for Training.
M Agus, E Gobbetti, G Pintore, G Zanetti, A Zorcolo
Eurographics Italian Chapter Conference 6, 183-187
Author Response
Thank you very much for your precious comments. We have modified our manuscript in accordance with your comments.
Comment 1_1: The first sentence in the abstract is not clear. In general what are the main motivations, and the actual contributions of the paper? It is not really clear from the abstract.
Answer 1_1: Thank you for your comment. Since part of the sentence following “but” is especially unclear, we have modified the sentences of that section. Please see lines 13 through 17 on page 1 of 19. In addition, we expect that our method will contribute to the standardization of surgical skill levels, becoming one of the basic approaches. Please see lines 28 through 30 on page 1 of 19
Comment 1_2: I also don't like to read in the abstract quantitative measures without a context and references.
Answer 1_2: Thank you for your comment. We have added a description briefly explaining AUC and IoU. Please see lines 22 through 25 on page 1 of 19.
The abstract, as modified in accordance with comments 1_1 and 1_2, is as follows
“Surgical skill levels of young ophthalmologists tend to be instinctively judged by ophthalmologists in practice, and hence a stable evaluation is not always made for a single ophthalmologist. Although it has been said that standardizing skill levels presents difficulty as surgical methods vary greatly, approaches based on machine learning seem to be promising for this objective. In this study, we propose a method for displaying the information necessary to quantify the surgical techniques of cataract surgery in real-time. The proposed method consists of two steps. First, we use InceptionV3, an image classification network, to extract important surgical phases and to detect surgical problems. Next, one of the segmentation networks, scSE-FC-DenseNet, is used to detect the cornea and the tip of the surgical instrument and the incisional site in the continuous curvilinear capsulorrhexis, a particularly important phase in cataract surgery. The first and second steps are evaluated in terms of the area under curve (i.e., AUC ) of the figure of the true positive rate versus (1false positive rate) and the intersection over union (i.e., IoU) obtained by the ground truth and prediction associated with the region of interest. As a result, in the first step, the network was able to detect surgical problems with an AUC of 0.97. In the second step, the detection rate of the cornea was 99.7% when the IoU was 0.8 or more, and the detection rates of the tips of the forceps and the incisional site were 86.9% and 94.9% when the IoU was 0.1 or more, respectively. It was thus expected that the proposed method is one of the basic techniques to achieve the standardization of surgical skill levels.”
Comment 1_3: I think also that the organization of the paper would improve if the discussion of related work is separated from the introduction. To this end, it would be useful to add a discussion about surgical training in cataract extraction. For an overview, authors can consider the papers:
Real Time Simulation of Phaco-emulsification for Cataract Surgery Training.
M Agus, E Gobbetti, G Pintore, G Zanetti, A Zorcolo
VRIPHYS, 91-100
Real-time Cataract Surgery Simulation for Training.
M Agus, E Gobbetti, G Pintore, G Zanetti, A Zorcolo
Eurographics Italian Chapter Conference 6, 183-187
Answer 1_3: Thank you for your comment. We have moved the sentences associated with related works to Section 2. In addition, we have added sentences associated with the papers that you mentioned at the end of Section 2, and have added those papers to reference list.
Reviewer 2 Report
The current manuscripts reports on the development and training of a neuronal network which is capable to detect problems in cataract surgeries even before the ophthalmologist can detect. The authors further claim a real time detection of the cornea and the surgical instrument. The manuscript is written very technically and the clinical relevance is missing in the text. Furthermore, the discussion part is weak and the conclusion is more a summary of the method and the results than a conclusion.
The idea of the manuscript is very interesting and could lead to great training tools. I miss information on the labeling and comparisons to other approaches.
Introduction:
- Remove wording in the introduction like "Reference 7 and 8 ..." Better: Sakabe et al (2019)
Methods:
- How was the labeling done? To which extent was the labeling done? What are the label classes?
- Why was the surgical instrument detction only done in no problem CCC phase? Does this mean that the problem detection does not work in presence of the instrument, or vise versa?
- How was the segmentation of the surgical instrument for labeling purposes perfomed? Was this an automized procedure or done by an expert?
- Figure capations needs be reworked
- What was the rational behind using Inception V3? Did the authors test also other models for their purposes? Same question for scSE-FC-DenseNet?
- Figure 5 and 6 are not needed to my opinion
- What hardware was used for training?
Results:
- What is the time unit for figure 8? Please add
- I dont understand the sense of Figure 11. It should explain the segmentations results and their viszualization for the surgical instrument. However, there are only overlapping squares shown. Would be better if realy images with ground truth and segmentation results are shown
- Please decribe the meaning of N in table 9 also in the capation and in the text
The discussion part is weak
- The authors mention that it is necessary the refine the proposed method regarding the time estimation of problem occurence. This is crucial. What do the authors mean here? It was further not clear from the methods section, what is a "problem". If the labeling was not created on basis of a clincial problem definition, the classification of the network does not make sense, of course
- The authors state that it would be nessary to introduce image processing. Why this was not done already for the current work?
Author Response
Thank you very much for your precious comments. We have modified our manuscript in accordance with your comments.
Comment 2_1: Remove wording in the introduction like "Reference 7 and 8 ..." Better: Sakabe et al (2019)
Answer 2_1: Thank you for your comment. We have changed the wording according to the above comment. Please see the words in lines 64 through 72 on page 2 of 19.
Comment 2_2: How was the labeling done? To which extent was the labeling done? What are the label classes?
Answer 2_2: Thank you very much for your comment. Ophthalmologists working at Tsukazaki Hospital watched video recordings of cataract surgeries, and checked time points when CCC started, nuclear extraction finished, and surgical problems occurred. Such time points were registered on electronic files. The files were used to annotate surgical phases in videos with labels. The number of classes is two. In other words, the label “important phase” is assigned to frames corresponding to the period between the two time points, from the time when CCC started through when nuclear extraction finished, while the label “other” is assigned to frames corresponding to the period exclusively included as time slots when neither CCC nor nuclear extraction were performed. In addition to annotating each frame as an important phase or other phase, please note that annotation for surgical problem detection requires the following two label classes: problem occurrence and no problem. It is performed according to electronic files with information associated with time points when surgical problems occurred. We have incorporated the above into Section 3. Please see the sentences in lines 98 through 101 on page 3 of 19 as follows.
“Ophthalmologists working at Saneikai Tsukazaki Hospital (Himeji, Japan) watched video recordings of cataract surgeries performed at the hospital, and checked the time points when CCC started, nuclear extraction finished, and surgical problems occurred. Such time points were registered on electronic files. The use of these video recordings has been approved by the Ethics Committee of Tsukazaki Hospital.”
Comment 2_3: Why was the surgical instrument detction only done in no problem CCC phase?
Answer 2_3: Thank you for your comment. The occurrence of a problem in a cataract surgery often results in either a low rate of progress or in interruption. The surgery then tends to consume much time. In such cases, no remarkable change appears between the consecutive frames of the video in which the surgery is recorded. When employing machine learning, acquiring data with high diversity is preferable to acquiring a large amount of data with low diversity. In other words, training data used to construct discrimination models by machine learning should be prepared from frames with clear changes made by surgical instruments. This is why surgical instrument detection is applied only to cases where no problems occur. We have added the above to Section 3. Please see the sentences in lines 126 through 133 on page 3 of 19 as follows.
“ Recall that the above electronic files have information on the occurrence of surgical problems. In addition to one of the two labels, important phase or other phase, a label to indicate the presence or absence of a surgical problem was also added to each video frame. The labeling was performed based on information associated with surgical problems in the above files, and it is safely said that the annotation of surgical problems was made by ophthalmologists working at Tsukazaki Hospital. The phase breakdown of the obtained image data is shown in Table 1, and a sample of the actual images of each phase are shown in Figure 1. Additionally, a problem breakdown is tabulated in Table 2.”
Comment 2_4: Does this mean that the problem detection does not work in presence of the instrument, or vise versa? “
Answer 2_4: Thank you for your comment. Problem detection can be applied to a video consisting of consecutive frames without surgical instruments, because the eye area is targeted for training in addition to the instruments. The accuracy of problem detection however decreases in such cases. We have added the description associated with the above to Section 6. Please see the sentences in lines 285 through 288 on page 9 of 19. as follows.
“Note that the results were obtained by applying the proposed method to videos with frames in which surgical instruments appeared. The proposed phase recognition and problem detection can be applied to a video consisting of consecutive frames without surgical instruments, because in addition to the instruments, the eye area is also targeted for training. However, their accuracy decreases in such cases.”
Comment 2_5: How was the segmentation of the surgical instrument for labeling purposes perfomed? Was this an automized procedure or done by an expert?
Answer 2_5: Thank you for your comment. Orthoptists working at Tsukazaki Hospital annotated surgical phases in videos with labels. They then used the annotation tool known as LabelMe. We have incorporated this description into Section 3. Please see the sentences in lines 138 through 139 on page 3 of 19 as follows. In addition, we have added URL address for LabelMe to the reference list. Please see Reference [15] on page 17 of 19.
“Ophthalmologists working at Tsukazaki Hospital annotated surgical instruments in videos with labels. They then used the annotation tool known as LabelMe [15].”
Comment 2_6: Figure capations needs be reworked
Answer 2_6: Thank you for your comment.
We have changed captions as follows.
Figure 1: We have added the following words, which might be recognized as part of main text, to its caption.
[A] CCC (Inamura forceps, retro illumination method), [B] nuclear extraction, [C] others (intraocular lens insertion)
Figure 2: We have added the following words, which might be recognized as part of main text, to its caption. Please note that the words “ground truth of” is added to [B], [C], and [D].
[A] input image, [B] ground truth of corneal area, [C] ground truth of forceps' tips, [D] ground truth of incisional site.
Figure 3: We have added some sentences explaining “Base,” “conv,” pooling layer, “Max Pooling,” “Average Pooling,” and “Filter Concat.” as follows. Please note that similar explanations appear in the main text.
"Base" means the input tensor to the Inception module. The "conv" means a convolutional layer at which the convolution is performed for a local region of the image, and learns high-order features of the image. The pooling layer compresses the tensors to reduce the amount of computation and prevent over-learning. "Max Pooling" computes the maximum value of the local region of the tensor, and "Average Pooling" computes the average value and compresses it. "Filter Concat" represents the concatenation of tensors.
Figure 5 (Figure 7 in the original manuscript): We have added the following words.
This evaluation is based on the risk level Dt.
Figure 6 (Figure 8 in the original manuscript): We have added sentences explaining the meaning of “0” and that of negative values to its caption as follows.
The "0" time point on the horizontal axis means that there was no difference between the problem-occurring time determined by the ophthalmologist and that detected by the NN. If the time is negative, it means that the NN detected the problem earlier than the ophthalmologist.
Figure 7 (Figure 9 in the original manuscript): We have added the words, which might be recognized as part of main text, to its caption as follows.
[A] the risk level for videos without problems, [B] the risk level for videos with problems.
Figure 8 (Figure 10 in the original manuscript): We have added the explanation on IoU to its caption as follows.
IoU is specified by the ground truth and prediction associated with the region of interest, and takes a value in the range of 0 to 1.
Figure 9 (Figure 11 in the original manuscript): We have added the explanation on certainty of segmentation, gradation map and so forth to its caption as follows.
NN outputs the certainty of segmentation for each pixel in the range of 0 to 1. The certainty that is closer to 1 indicates a class that NN wants to detect. Images [B] – [D] are overlapping image [A] with gradation maps, showing blue when the certainty value is 0, and red when the certainty value is 1. Image [E] shows a circumscribed circle of the segmented figure by binarizing with the certainty of 0.5 as a threshold value. In addition, the ground truth is the same as in Figure 2.
Figure 11 (Figure 13 in the original manuscript): We have added the sentence on the order of CCC phase to its caption as follows.
The cataract surgery pcoceeds in the order of images [A] to [E].
Table 6 (Table 5 in the original manuscript): We have changed some words to make the meaning clearer as follows.
Types of image processing used for learning corneal and positions of surgical instruments, and their parameters.
Table 10 (Table 9 in the original manuscript): We have added the sentences explaining the meaning of IoU³N where N is a parameter to its caption as follows.
N is a parameter related to IoU. IoU>N means that the ratio of area obtained by overlapping the prediction result with ground truth compared with the area obtained by uniting the former to the latter is larger than or equal to N.
Comment 2_7: What was the rational behind using Inception V3? Did the authors test also other models for their purposes? Same question for scSE-FC-DenseNet?
Answer 2_7: Thank you for your comment. When we determined NN for recognizing cataract surgery phases and detecting surgical problems, we placed importance on the trade-off between the time required for classification and classification accuracy. We determined to employ InceptionV3 on the basis of the following paper: Simone et al, “Benchmark Analysis of Representative Deep Neural Network Architectures,” IEEE Access (Volume: 6), 2018. We have added the explanation associated with the above and this paper. Please see lines 159 through 163 on page 4 through 5 of 19 and Reference [16] as follows.
“We selected InceptionV3, which provides high performance with real-time operating capability, by referring to the benchmark [16] which investigated the relationship between the performance of major NN and computational capacity.”
Next, let us explain scSE-FC-DenseNet. High computational complexity is imposed on a segmentation NN, and hence the NN must be carefully designed if its response time is shortened so that it can be safely said to be nearly real time NN. We determined to employ scSE-FC-DenseNet on the basis of the following paper: Roy et al, “Concurrent Spatial and Channel Squeeze & Excitation in Fully Convolutional Networks.” MICCAI, 2018. In this paper, scSE-FC-DenseNet is considered to be suitable to adjust the computational complexity imposed on it. We have added the explanation associated with the above and this paper. Please see lines 194 through 200 on page 6 of 19 and Reference [17, 20] as follows. We have not tried to apply other NNs.
“In this study, one of the segmentation NNs, scSE-FC-DenseNet, was used to detect the corneal area of the patient and track the surgical instruments. High computational complexity is imposed on a segmentation NN, and hence the NN must be carefully designed if its response time is shortened so that the NN can safely be said to be nearly real time. The scSE-FC-DenseNet is an FC-DenseNet [19], which incorporates the Dense block proposed in DenseNet [17] into U-Net [18], with an attention mechanism called the scSE (Spatial and Channel Squeeze & Excitation) module [20]. It is discussed in [19] that employing DenseNet enables us to easily adjust the computational complexity imposed on the NN.”
Comment 2_8: Figure 5 and 6 are not needed to my opinion
Answer 2_8: Thank you for your comment. We agree with your opinion, and have deleted Figs. 5 and 6 in the original manuscript (new figure numbering has been applied in this resubmission). In addition, descriptions in the text, which are related to the deleted Figures 5 and 6, were also deleted. Please see lines 206 through 214 on page 6 of 19 as follows.
“DenseNet is a NN that uses a structure called Dense block, which combines a skipped connection and a bottleneck layer. This structure allows us to construct NNs that do not increase the number of parameters explosively, even when the convolutional layer is more multi-layered. FC-DenseNet is a NN that replaces the usual convolutional layer used in U-Net with this Dense block.
The scSE module is a combination of the Spatial Squeeze and Excitation (cSE) proposed in SE-Net, an image classification NN [21], which averages the whole image for each channel (Squeeze) and gives its attention (Excitation), and an sSE (Channel Squeeze and Spatial Excitation) that squeezes to the channel direction and excites each pixel. This module can effectively introduce the attention mechanism in a segmentation NN.”
Comment 2_9: What hardware was used for training?
Answer 2_9: Thank you for your comment. The network was trained on a system with two NVIDIA GTX 1080 Ti GPUs and the evaluation was done on a system with a single GPU. We have added a description associated with this sentence. Please see lines 235 through 236 on page 7 of 19 and 273 through 274 on page 8 of 19 as follows.
“The network was trained on a system with two NVIDIA GTX 1080 Ti GPUs and the evaluation was done on a single GPU.”
“The network was trained on a system with two NVIDIA GTX 1080 Ti GPUs and the evaluation was done on a single GPU.”
Comment 2_10: What is the time unit for figure 8? Please add.
Answer 2_10: Thank you for your comment. Unit is seconds. We have added it. Please see Figure 6 (Figure 8 in the original manuscript) as follows.
“The "0" time point on the horizontal axis means that there was no difference between the problem-occurring time determined by the ophthalmologist and that detected by the NN. If the time is negative, it means that the NN detected the problem earlier than the ophthalmologist.”
Comment 2_11: I dont understand the sense of Figure 11 (now Figure 9 with new numbering). It should explain the segmentations results and their viszualization for the surgical instrument. However, there are only overlapping squares shown. Would be better if realy images with ground truth and segmentation results are shown
Answer 2_11: Thank you for your comment. The trained scSE-FC-DenseNet40 outputs a value belonging to the range 0 to 1 as certainty values of segmentation for each pixel. It is judged that the segmentation result is appropriate as its certainty approaches the value 1. The certainty values of segmentation for each pixel are made into a gradation map. The maps overlap with images to be segmented as shown in Figures 9 [B]-[D]. Here, the color becomes bluer (or redder) as certainty of segmentation for each pixel approaches the value 0 (or 1). Binarization is executed provided that the threshold certainty of segmentation for each pixel is set to the value 0.5. The circumscribed circle is next depicted for each of the segmented objects. Figure 9 [E] is then obtained as a result. Note that its ground truth is depicted as shown in Figure 2. We have added the above to the end of Section 6. Please review.
Comment 2_12: Please decribe the meaning of N in table 9 also in the capation and in the text
Answer 2_12: Thank you for your comment. The value of N is a parameter related to IoU in equation (5). The maximum of IoU is equal to 1, and this means the prediction result obtained by the proposed method perfectly matches to the ground truth. As a result, IoU ³ N means that the ratio of area obtained by overlapping the prediction result with ground truth compared with area obtained by uniting the former to the latter is larger than or equal to N. The areas of the incisional site and of the tips of the forceps are comparatively small. Even in the case of N=0.1, therefore, it seems that results for tracing the incisional site and the tips of the forceps causes no major problems during practical use. We have added sentences associated with the above to the caption, text, and formula. Please see the caption of Table 10, lines 330 through 334 on page 12 of 19 as follows, and the equation (6).
“IoU takes the value 1 as its maximum. IoU=1 means the prediction result obtained by the proposed method perfectly matches the ground truth. means that the ratio of the area obtained by overlapping the prediction result with ground truth compared with the area obtained by uniting the former to the latter is larger than or equal to N. The correct response rate for the cornea showed remarkably high accuracy. On the other hand, the correct response rates for the incisional site and the tips of the forceps were 94.9% and 86.9% respectively, when the value of N in the equation (6) was 0.1. In other words, the value obtained with the equation (6) is not high under N = 0.8. However, as shown in Figure 2, the area for which the correct label is assigned for surgical instrument detection is exceedingly small; therefore, even with the criteria, it seems that results for tracing the incisional site and the tips of forceps causes no major troubles during practical use. It can be thus considered that the NN is capable of tracking instruments successfully.”
Comment 2_13: The discussion part is weak
Answer 2_13: Thank you for your comment. We have added description on potential of the proposed method. Please see lines 401 through 412 on page 14 of 19 as follows.
Comment 2_14: The authors mention that it is necessary the refine the proposed method regarding the time estimation of problem occurence. This is crucial. What do the authors mean here?
Answer 2_14: Thank you for your comment. In some videos to be checked, there are cases where large differences arise between the problem-occurring time determined by the proposed method and that the time given by the ophthalmologists. It seems that this is caused by differences that sometimes arise in evaluation criteria for surgical techniques between ophthalmologists and the proposed method. The proposed method should be based on evaluation criteria for surgical techniques that the majority of ophthalmologists consider appropriate. To achieve this objective, it is unfavorable that differences sometimes arise in the evaluation criteria. The proposed method therefore must be refined so that its criteria can be as close as possible to those of the ophthalmologists. We have added sentences associated with the above to Section 7. Please see lines 373 through 389 on page 13 through 14 of 19.
Comment 2_15: It was further not clear from the methods section, what is a "problem". If the labeling was not created on basis of a clinical problem definition, the classification of the network does not make sense, of course
Answer 2_15: Thank you for your comment. We have added Table 2 showing the types of problems. Please review.
Comment 2_16: The authors state that it would be necessary to introduce image processing. Why this was not done already for the current work?
Answer 2_16: Thank you for your comment. We have tried to apply several schemes of general image processing. An essential scheme has not yet been found. It is possible that engaging in schemes specialized for cataract surgery is preferable to examining well-known schemes. We will further search for such an image processing scheme. We have added sentences associated with the above to Section 7. Please see lines 395 through 400 on page 14 of 19.
Round 2
Reviewer 1 Report
The paper greatly improved since original submission.
To my opinion, it is in good shape for publication. I just recommend authors to double check for language issues, if possible with the help of a native speaker.